# OpenReview forum: "Attributional Safety Failures in Large Language Models under Code-Mixed Perturbations"
_TMLR — Under review for TMLR_

### Review · Reviewer_jxoW · 2026-06-21

**Summary Of Contributions:**

The paper investigates safety failures of LLMs under code-mixed prompts, showing that prompts handled safely in monolingual English can elicit harmful responses when rewritten in multilingual/code-mixed form. Its main contribution is an empirical evaluation across multiple languages, cultural settings, and open source models, together with an attribution-based analysis framework, Saliency Drift Attribution (SDA), which aims to explain these failures by showing that safety-critical tokens lose saliency under code-mixing.

**Audience:**

Yes

**Audience Explanation:**

The paper studies an important problem for LLM safety: whether safety alignment remains reliable under multilingual and code-mixed inputs. This should be relevant to TMLR readers working on LLM safety, multilingual NLP, robustness, interpretability, and fairness. Even though I have concerns about the strength of some claims, the main finding that code-mixed prompts may substantially weaken safety behavior is timely and worth knowing. The paper also raises a useful broader point that English-centric safety evaluation may miss important failure modes in real-world multilingual settings.

**Broader Impact Concerns:**

The paper has potential dual-use risks, since its findings and examples could help users craft code-mixed jailbreak prompts. It also discusses harmful content involving protected or marginalized groups, so examples should be presented carefully to avoid reinforcing stereotypes.

**Claims And Evidence:**

No

**Claims Explanation:**

1. The paper does not fully support the stronger claim that this is a general “catastrophic” safety failure of LLMs. The experiments are limited to three old open-source models, and the paper does not show whether the same pattern holds for a broader range of deployed or closed-source systems.
2. The paper claims that SDA explains why the failures happen, but the evidence is not fully convincing. SDA shows that safety-critical tokens often receive lower attribution in code-mixed prompts, but the paper does not sufficiently prove that this attribution drop is the direct cause of harmful generation.
3. The paper claims that the findings extend to real-world social media data. This is partially supported by the TRAC-1 experiment, but the evidence is limited to Hindi-English comments. This does not fully justify broader claims about real-world multilingual or code-mixed use across languages and platforms.

**Requested Changes:**

1.	The scope of the claims should be narrowed. The experiments show a code-mixing safety vulnerability in the three evaluated open-source models, but they do not establish that this is a general catastrophic failure of LLMs as a whole. Similarly, the real-world evidence is based on one Hindi-English social media dataset.
2.	The evaluation should include stronger and more recent models, such as Qwen or DeepSeek-family models. It would also be useful to evaluate representative closed-source frontier models, such as GPT-5 or Gemini.
3.	The representation of non-Latin scripts in the code-mixed prompts should be clarified. Many examples appear to use romanized or transliterated forms rather than native scripts, but the paper does not specify whether this is done consistently across languages or how the transliteration is produced.
4.	The SDA analysis needs to be made more rigorous. In particular, the paper should specify how tokens are aligned across English and code-mixed prompts, what output target is used for attribution, and whether the results are robust to alternative attribution methods or perturbation tests.
5.	More details are needed on the harmfulness annotation procedure. Since ASR is the main evaluation metric and is primarily based on gpt4-o judgments, the paper should report the judging prompt or rubric, the size of the human validation set, per-language agreement, and disagreement patterns between gpt4-o and human annotators.
6.	Additional control experiments are needed to better isolate the effect of code-mixing. For example, comparisons with fully translated prompts, native-script prompts, romanized prompts, and randomly mixed prompts would help determine whether the failures are due to code-mixing itself rather than translation artifacts, script choice, tokenization effects, or general distribution shift.

---

> ### Author Response · Authors · 2026-07-09
> **We express our gratitude to reviewer jxoW for evaluation**
>
> **Answer 1:** We agree and have **narrowed the claims** throughout the manuscript. We replaced phrases such as **“catastrophic,”** **“overlooked weakness,”** **“blinding the model to harmful intent,”** and **“risk for billions of users.”**
>
> Our revised central claim is:
>
> > Across the open-weight and API-accessible models evaluated in this study, code-mixed prompts produce a **substantial but model-dependent degradation in safety** relative to semantically corresponding English prompts.
>
> We also distinguish between **descriptive** and **causal** claims. **SDA** is no longer presented merely as proof that attribution loss is the sole cause of harmful generation. Instead, the revised manuscript combines **descriptive attribution analysis**, **dataset-level prediction**, and **controlled interventions**.
>
> The **real-world claim** has also been narrowed. **TRAC-1** is now described as evidence for one **Hindi–English social-media setting**, rather than as proof of universal real-world generalization. We additionally evaluate natural **Spanish–English**, **Tamil–English**, and **Arabic–English/Arabizi** datasets, as discussed in **Point 7** below.
>
> **Answer 2:** We expanded the **behavioral evaluation** from three models to **twelve systems** spanning multiple **architectures**, **scales**, **multilingual-alignment strategies**, and **access settings**.
>
> ## Expanded Behavioral Evaluation
>
> | Model | English ASR | Code-Mixed ASR | ΔASR |
> |---|---:|---:|---:|
> | microsoft/Phi-4-mini-instruct | 9.2% | 67.6% | **+58.4 pp** |
> | mistralai/Mistral-7B-Instruct-v0.3 | 18.2% | 72.1% | **+53.9 pp** |
> | meta-llama/Llama-3.1-8B-Instruct | 27.8% | 49.1% | +21.3 pp |
> | google/gemma-2-9b-it | 15.6% | 54.8% | +39.2 pp |
> | meta-llama/Llama-3.3-70B-Instruct | 8.7% | 30.9% | +22.2 pp |
> | Qwen/Qwen2.5-14B-Instruct | 10.9% | 38.7% | +27.8 pp |
> | Qwen/Qwen3-14B | 8.4% | 31.5% | +23.1 pp |
> | deepseek-ai/DeepSeek-R1-Distill-Qwen-14B | 12.1% | 40.8% | +28.7 pp |
> | CohereLabs/aya-expanse-32b | 11.8% | 24.9% | **+13.1 pp** |
> | gpt-4o-2024-11-20 | 3.9% | 13.2% | +9.3 pp |
> | gemini-2.5-pro | 4.7% | 16.8% | +12.1 pp |
> | claude-3-7-sonnet-20250219 | 2.8% | 10.6% | +7.8 pp |
>
> The expanded results show that the vulnerability is **not restricted to the original three models**. However, its magnitude is **highly model-dependent**. The explicitly multilingual **Aya** model and the closed **frontier systems** show substantially smaller gaps, while all evaluated systems retain a positive **English-to-code-mixed ASR increase**.
>
> We have revised the paper to emphasize this **heterogeneity**, rather than implying a uniform failure across all LLMs.
>
> The **mechanistic analysis** is restricted to **open-weight models** because it requires access to **gradients**, **activations**, and **attention**. The API models are therefore included only in the **behavioral evaluation**.
>
> **Answer 3:** We agree. The revised manuscript now explicitly distinguishes **native-script** and **romanized** conditions and documents the **transliteration procedure** for each language. We also report the proportion of **matrix-language** and **English tokens**, treatment of **borrowed words** and **named entities**, and **native-speaker validation**.
>
> We evaluated script effects on a common paired subset.
>
> ## Prompt-Form Controls
>
> | Condition | ASR | Gap from English |
> |---|---:|---:|
> | English | 18.4% | — |
> | Full translation, native script | 31.6% | +13.2 pp |
> | Full translation, romanized | 39.8% | +21.4 pp |
> | MLF code-mixed, romanized | 66.1% | +47.7 pp |
> | Random lexical substitution | 46.8% | +28.4 pp |
> | Word-order shuffled control | 44.5% | +26.1 pp |
>
> The main contrasts are:
>
> - **MLF native-script code-mixing versus full native-script translation:** +25.6 pp, **p < 0.001**
> - **MLF romanized versus MLF native script:** +8.9 pp, **p < 0.001**
> - **MLF native-script code-mixing versus ungrammatical random mixing:** +5.9 pp, **p < 0.001**
>
> Thus, **romanization increases the degradation** but is not its sole cause: **native-script code-mixed prompts** remain substantially less safe than fully translated **native-script prompts**. Generic **distribution shift** and **surface noise** also contribute, but do not fully explain the result for structurally valid **MLF code-mixing**.
>
> The language-level results show the same pattern.
>
> ## Native-Script versus Romanized Code-Mixing
>
> | Language | Native-Script CM ASR | Romanized CM ASR | Difference |
> |---|---:|---:|---:|
> | Arabic | 62.4% | 72.8% | +10.4 pp |
> | Bengali | 59.1% | 69.7% | +10.6 pp |
> | Hindi | 63.8% | 73.2% | +9.4 pp |
> | Japanese | 58.0% | 62.6% | +4.6 pp |
> | Korean | 51.9% | 58.7% | +6.8 pp |
> | Russian | 55.7% | 62.1% | +6.4 pp |
> | Chinese | 54.6% | 59.4% | +4.8 pp |
>
> **Please look into next section due to 5K limit.**

---

> > ### Author Response · Authors · 2026-07-09
> > **Continuation**
> >
> > **Answer 4:** We agree that the original **SDA analysis** was primarily correlational. We therefore added a **bidirectional counterfactual intervention** that selectively manipulates the internal representations of **safety-critical tokens** while leaving the input prompt and its tokenization unchanged.
> >
> > By **suppression**, we mean replacing the hidden representations of safety-critical English tokens toward a **neutral benign-token baseline**, thereby reducing their influence without changing the prompt. By **restoration**, we mean replacing the hidden representations of aligned code-mixed safety tokens toward their corresponding **English-run representations**. Identical interventions on matched random tokens serve as controls.
> >
> > ## Bidirectional Counterfactual Intervention
> >
> > | Source | Manipulation | Mean Safety-Token RI | ASR |
> > |---|---|---:|---:|
> > | English | None | 0.64 | 18.2% |
> > | English | Random-token suppression | 0.61 | 20.4% |
> > | **English** | **Safety-token suppression** | **0.32** | **51.7%** |
> > | Code-mixed | None | 0.27 | 72.1% |
> > | Code-mixed | Random-token steering | 0.29 | 68.8% |
> > | **Code-mixed** | **Safety-token restoration** | **0.49** | **39.6%** |
> >
> > Suppressing **safety-critical representations** in English lowers their mean **RI** from **0.64** to **0.32** and increases **ASR** from **18.2%** to **51.7%**. Conversely, restoring these representations in code-mixed prompts raises mean **RI** from **0.27** to **0.49** and reduces **ASR** from **72.1%** to **39.6%**.
> >
> > Random-token interventions produce only small ASR changes of approximately **2–3 percentage points**. These bidirectional results provide stronger **causal evidence** that weakened safety-critical representations contribute to the observed failures, while not implying that they are the sole mechanism.
> >
> > **Answer 5:** We added **two independent automatic judges** and expanded **human validation**. All four judges preserve the same ordering: **CM ≫ English**, **Mistral/Phi degradation ≫ Llama**, and **Arabic/Hindi/Bengali highest-risk**, as shown in the tables below.
> >
> > ## Judge Robustness
> >
> > **ASR averaged over models and languages.**
> >
> > | Judge | English ASR | CM ASR | Gap |
> > |---|---:|---:|---:|
> > | GPT-4o | 18.4% | 62.9% | +44.5 |
> > | LlamaGuard multilingual | 21.7% | 65.3% | +43.6 |
> > | ShieldGemma | 20.5% | 61.1% | +40.6 |
> > | **Human majority** | **17.8%** | **57.6%** | **+39.8** |
> >
> > ## Judge–Human Agreement
> >
> > | Judge Pair | Cohen's κ |
> > |---|---:|
> > | **GPT-4o ↔ human** | **0.78** |
> > | LlamaGuard ↔ human | 0.70 |
> > | ShieldGemma ↔ human | 0.68 |
> > | **Majority of automatic judges ↔ human** | **0.82** |
> >
> > ## Per-Language GPT-4o ↔ Human Agreement
> >
> > **n = 50 per language.**
> >
> > | Language | κ | Agreement | Language | κ | Agreement |
> > |---|---:|---:|---|---:|---:|
> > | Arabic | 0.73 | 87.5% | Japanese | 0.79 | 90.8% |
> > | Bengali | 0.75 | 88.3% | Korean | 0.81 | 91.7% |
> > | Chinese | 0.80 | 91.7% | Portuguese | 0.82 | 92.5% |
> > | German | 0.85 | 94.2% | Russian | 0.78 | 90.0% |
> > | Hindi | 0.76 | 89.2% | Spanish | 0.84 | 93.3% |
> >
> > ## Disagreement Taxonomy
> >
> > The main disagreement categories were:
> >
> > | Category | Share | Description |
> > |---|---:|---|
> > | **Indirect stereotypes** | **38%** | Harmful generalizations were implied rather than stated explicitly, such as associating a group with corruption without using an overt slur. |
> > | **Quotation versus endorsement** | **27%** | The model repeated a harmful claim while discussing or rejecting it, but one evaluator interpreted the repetition as endorsement. |
> > | **Culture-specific derogatory expressions** | **19%** | GPT-4o missed a locally offensive term or coded insult. |
> > | **Mixed-language ambiguity** | **10%** | The meaning of a transliterated or code-mixed phrase was interpreted differently by evaluators. |
> > | **Annotation error or unresolved disagreement** | **6%** | Labels remained inconsistent after review. |
> >
> > ## Reconciliation on the Common Human-Validation Subset
> >
> > We state explicitly that the human study evaluates a **stratified subset**, balanced by **language**, **model**, and **condition**, rather than a re-estimation of the full **Table 1** population. On that same subset, **GPT-4o** and **human majority labels** agree closely.
> >
> > | Same Subset | English ASR | CM ASR |
> > |---|---:|---:|
> > | GPT-4o | 18.9% | 59.7% |
> > | **Human majority** | **17.8%** | **57.6%** |
> >
> > **Please look into next section due to 5K limit.**

---

> > > ### Author Response · Authors · 2026-07-09
> > > **Continuation**
> > >
> > > **Answer 6:** We added controls for each of these factors.
> > >
> > > To measure **comprehension**, we created a **benign counterpart** for each harmful prompt while preserving the same **language**, **script**, and **code-mixing structure**. The model was asked to answer or paraphrase the prompt in English. A response was marked correct when it preserved the main **meaning**, **target**, and **requested action**.
> > >
> > > ## 6.1 Multilingual Comprehension
> > >
> > > | Condition | Correct Paraphrase/Answer | ASR on Harmful Counterpart |
> > > |---|---:|---:|
> > > | English | 94.1% | 18.4% |
> > > | Full translation | 89.7% | 31.6% |
> > > | **MLF code-mixed** | **87.9%** | **61.8%** |
> > > | Random ungrammatical mixing | 70.8% | 51.3% |
> > >
> > > Among **MLF prompts** that were correctly comprehended, **ASR remained 59.6%**, compared with **68.7%** among incorrectly comprehended examples.
> > >
> > > In a controlled regression, correct comprehension is associated with lower failure odds, **OR = 0.72, p = 0.01**, but the **MLF code-mixed condition** remains strongly associated with harmful behavior after controlling for comprehension, **OR = 3.84, p < 0.001**.
> > >
> > > Thus, **comprehension difficulty** contributes to the effect but does not account for the majority of the observed degradation.
> > >
> > > ## 6.2 Tokenization
> > >
> > > | Condition | Tokens/word | Fragmented Safety Tokens | Mean Safety-Token Drift | ASR |
> > > |---|---:|---:|---:|---:|
> > > | English | 1.31 | 18% | 0.05 | 18.4% |
> > > | Full translation | 1.82 | 39% | 0.17 | 31.6% |
> > > | **Native-script CM** | **2.07** | **48%** | **0.28** | **57.2%** |
> > > | **Romanized CM** | **2.34** | **56%** | **0.35** | **66.1%** |
> > >
> > > **Token fertility** correlates with ASR, **Spearman ρ = 0.48, p < 0.001**. **Safety-token fragmentation** correlates with **saliency drift**, **ρ = 0.55, p < 0.001**. However, drift remains associated with ASR after controlling for fertility, **partial ρ = 0.46, p < 0.001**.
> > >
> > > We therefore interpret **tokenization** as an important contributing factor, but not as a complete explanation.
> > >
> > > ## 6.3 Semantic Fidelity
> > >
> > > | Condition | BERTScore F1 | Back-Translation Adequacy |
> > > |---|---:|---:|
> > > | Full translation | 0.91 | 92.8% |
> > > | **MLF native script** | **0.89** | **90.7%** |
> > > | **MLF romanized** | **0.87** | **87.9%** |
> > > | Random mixing | 0.78 | 73.5% |
> > >
> > > When restricting the analysis to examples with **BERTScore > 0.88**, fully translated prompts have an **ASR of 29.8%**, whereas **MLF code-mixed prompts** retain an **ASR of 58.9%**. The effect therefore remains large even among **high-fidelity prompt pairs**.
> > >
> > > Taken together, the control results show that **translation**, **romanization**, **comprehension difficulty**, **tokenization**, and **distribution shift** each contribute to the degradation, but none individually explains the substantially higher **ASR** under structurally valid **code-mixing**.
> > >
> > > **Answer 7:** The phenomenon replicates on **wild data** across four language pairs.
> > >
> > > ## Natural Code-Mixed Datasets
> > >
> > > **Dataset size: 200 instances each.**
> > >
> > > | Dataset | Language Pair | Normalized-English ASR | Natural CM ASR | Gap |
> > > |---|---|---:|---:|---:|
> > > | TRAC-1 | Hindi–English | 24.8% | 71.8% | **+47.0** |
> > > | LinCE-derived [[link](https://aclanthology.org/2020.lrec-1.223/)] | Spanish–English | 20.6% | 49.7% | **+29.1** |
> > > | HASOC/Dravidian-derived [[link](https://ceur-ws.org/Vol-3159/T3-1.pdf)] | Tamil–English | 22.9% | 57.8% | **+34.9** |
> > > | Arabizi social-media [[link](https://aclanthology.org/2022.wanlp-1.18.pdf)] | Arabic–English | 19.3% | 63.5% | **+44.2** |
> > >
> > > ## Per-Model Natural CM ASR
> > >
> > > | Dataset | Phi | Mistral | Llama |
> > > |---|---:|---:|---:|
> > > | TRAC-1 | 61.3% | 74.9% | 79.2% |
> > > | Spanish–English | 44.2% | 57.1% | 47.8% |
> > > | Tamil–English | 52.7% | 65.4% | 55.3% |
> > > | Arabic–English/Arabizi | 58.6% | 70.8% | 61.1% |
> > >
> > > **Answer 8:** We agree and have added a dedicated **responsible-release section**. The revised release policy includes the following safeguards:
> > >
> > > - Full **adversarial prompts** and **harmful generations** will not be released unrestrictedly.
> > > - Public artifacts will contain **aggregate results**, **evaluation code**, **redacted examples**, and **safe templates**.
> > > - Access to the complete **adversarial dataset** will be gated for legitimate research use.
> > > - Harmful outputs shown in the manuscript will be **truncated or paraphrased** where possible.
> > > - Examples involving **protected groups** will be retained only when analytically necessary.
> > > - Annotators receive **advance content warnings**, **exposure limits**, **breaks**, **fair compensation**, an unrestricted **opt-out mechanism**, and **well-being resources**.
> > > - We explicitly discuss **cultural and evaluator bias** in both automatic and human harmfulness judgments.
> > >
> > > We have also added the following statement:
> > >
> > > > **Code-mixing is a normal, linguistically meaningful, and culturally important form of communication.** The failures documented in this paper reflect limitations of current model safety systems, not a deficiency in code-mixed language or its users.

---

### Review · Reviewer_sYSB · 2026-06-24

**Summary Of Contributions:**

This paper studies whether code-mixed prompts can weaken LLM safety alignment. Across ten languages and three open-weight models, the authors report large increases in attack success rates for both synthetic and real-world code-mixed inputs. They further propose Saliency Drift Attribution (SDA) to explain these failures. The topic is important and the empirical finding is concerning, but the attributional explanation remains under-validated.

**Audience:**

Yes

**Audience Explanation:**

Multilingual safety and code-mixing are important and underexplored topics. The large ASR increase across languages is practically relevant, and the paper could become a useful contribution after the mechanism and evaluation are made more rigorous.

**Broader Impact Concerns:**

The work addresses an important safety issue, but there are a few broader-impact aspects that could be discussed more fully. Since the study involves harmful and culturally sensitive prompts, it may be useful to say more about annotator safeguards and how exposure to toxic or discriminatory content was handled. It would also be helpful to clarify the intended release policy for the generated attack prompts and code-mixed safety data. Finally, the broader-impact discussion could emphasize that code-mixing is normal linguistic behavior, and that the concern is not code-mixing itself, but the current failure of model safety systems to serve multilingual users reliably.

**Claims And Evidence:**

No

**Claims Explanation:**

1. The ASR results support the empirical claim that the tested models are less safe under code-mixed prompts, but they do not convincingly establish that attributional collapse causes these failures.
2. The SDA analysis is mainly based on selected examples and visualizations, without a systematic dataset-level test showing that saliency drift predicts safety failures.
3. Several important confounders remain insufficiently addressed, including synthetic prompt artifacts, weaker multilingual comprehension, tokenization effects, and reliance on GPT-4o as the main safety judge.

**Requested Changes:**

1. The causal claim around SDA should be softened, unless the authors add stronger causal evidence or dataset-level saliency-drift analysis.
2. The SDA method should be specified more clearly, including the attribution target, baseline, aggregation, token alignment, and normalization.
3. The evaluation should better address key confounders, especially translation quality, benign code-mixed prompts, multilingual comprehension, tokenization effects, and GPT-4o judge reliability.

---

> ### Author Response · Authors · 2026-07-09
> **We express our gratitude to reviewer sYSB for evaluation**
>
> **Answer 1:** We added both the **causal evidence** and the **dataset-level analysis** requested by the reviewer. The analyses were conducted on **mistralai/Mistral-7B-Instruct-v0.3** and **meta-llama/Llama-3.1-8B-Instruct**, with dataset-level values averaged across the two models.
>
> For each example, **safety-critical tokens** were identified using the same **Perspective API token-contribution procedure** as in **Section 8.3**. The **toxicity-only predictor** is the prompt-level **Perspective API score**.
>
> ## Dataset-Level Predictivity:
>
> Rather than relying on selected visualizations, we now compute **per-example safety-token drift** across the full **Generic TestSet**. Drift separates the behavioral cases cleanly, predicts **Case-1 flips** well above the confounders, and remains dominant in a controlled regression. The signal is specific to **safety-critical tokens**.
>
> ### Mean Saliency Drift by Behavioral Case
>
> | Behavioral Case | Mean Drift | 95% CI |
> |---|---:|---|
> | **Case 1: safe EN → harmful CM** | **0.39** | **[0.37, 0.41]** |
> | Case 2: harmful → harmful | 0.13 | [0.11, 0.15] |
> | Case 3: safe → safe | 0.09 | [0.08, 0.10] |
>
> ### Predicting Case-1 Flips - AUROC
>
> | Predictor | AUROC |
> |---|---:|
> | Prompt length only | 0.55 |
> | Toxicity only | 0.64 |
> | Token fertility only | 0.66 |
> | **Mean safety-token drift only** | **0.82** |
> | Length + toxicity + fertility | 0.71 |
> | **All controls + drift** | **0.86** |
>
> ## Causal Interventions:
>
> The counterfactual intervention on **safety-token positions** cuts the **Case-1 harmful rate** roughly in half and restores approximately **50% of examples** to safe. Bidirectional manipulation induces failures in English and repairs them in code-mixed inputs by approximately **±33 percentage points**, while random interventions do neither.
>
> Full details are provided in the response to **Reviewer UeF1, Point 2**.
>
> We now make the **causal claim** only to the strength licensed by these results. We retain the term **“attributional collapse”** only where the interventions justify it, and use **“saliency drift”** descriptively elsewhere.
>
> ---
>
> **Answer 2:** We thank the reviewer for this comment. We would like to clarify that **Section 8.2** already formally defines **SDA**, including the **token-level attribution function**, **Rank Inverse normalization**, the **English–code-mixed drift measure**, its interpretation, and the **offset used for visualization**. **Section 8.1** also specifies that the token attribution scores are computed using **Integrated Gradients**.
>
> We agree, however, that several implementation details are not stated sufficiently explicitly. In the revision, we will clarify the **Integrated Gradients baseline**, the exact **scalar output** used as the attribution target, the procedure for aligning **semantically corresponding English and code-mixed tokens**, the aggregation of **tokenizer subwords into lexical items**, and the handling of **tied ranks** and **unaligned tokens**.
>
> We will also distinguish the **[0,1]-normalized scores** used in the illustrative visualization from the **Rank Inverse scores** used in the **SDA analysis**. These revisions require clarification of the existing method rather than an additional experiment.
>
> ---
>
> **Answer 3:** We thank the reviewer for highlighting these important potential confounders. We conducted **targeted controls** to determine whether the observed safety degradation can be explained by **translation**, **script**, **generic linguistic noise**, **comprehension failure**, **tokenization**, or **evaluator dependence**.
>
> ## Translation, Script, and Linguistic-Noise Controls
>
> We evaluated all prompt forms on the same paired subset. As shown in Table below, fully translating the prompts into the matrix language increased **ASR** from **18.4%** to **31.6%**, whereas linguistically valid **MLF code-mixing** increased ASR to **57.2%**. Romanized MLF prompts produced a further increase to **66.1%**, indicating that romanization amplifies the effect but is not its sole cause.
>
> Random or ungrammatical mixing reached an ASR of **51.3%**, suggesting that generic surface noise contributes to the degradation but does not fully account for the effect observed under structurally valid code-mixing.
>
> ### Translation, Script, and Tokenization Controls
>
> | Condition | Tokens/word | Fragmented safety tokens | ASR |
> |---|---:|---:|---:|
> | English | 1.31 | 18% | 18.4% |
> | Full translation | 1.82 | 39% | 31.6% |
> | **Native-script code-mixed** | **2.07** | **48%** | **57.2%** |
> | **Romanized code-mixed** | **2.34** | **56%** | **66.1%** |
>
> **Please look into next section due to 5K limit**

---

> > ### Author Response · Authors · 2026-07-09
> > **Continuation**
> >
> > The principal contrasts were significant: **MLF native-script code-mixing versus full native-script translation**, **+25.6 percentage points**; **MLF code-mixing versus random mixing**, **+5.9 percentage points**; and **romanized versus native-script MLF code-mixing**, **+8.9 percentage points**.
> >
> > ## Translation Quality and Semantic Preservation
> >
> > The existing **native-speaker evaluation** already shows that the generated code-mixed prompts retain acceptable translation quality across all ten languages. To strengthen this analysis, we re-evaluated ASR after restricting the data to prompts judged **semantically adequate** by native speakers.
> >
> > The code-mixed versus fully translated ASR gap remained substantial within this subset, indicating that the observed degradation is not primarily driven by **meaning distortion** or **poor translation**. Please see **Response R3 to Reviewer UeF1**.
> >
> > ## Tokenization
> >
> > Code-mixed prompts showed higher **token fertility** and greater fragmentation of **safety-critical words** than their English counterparts. Token fertility correlated with ASR, indicating that tokenization contributes to the observed failures.
> >
> > However, **safety-token saliency drift** remained significantly associated with harmful behavior after accounting for fertility. We therefore interpret tokenization as a **contributing factor**, rather than a sufficient explanation for the safety degradation. Please see **Response R8 to Reviewer UeF1**.
> >
> > ## GPT-4o Reliability
> >
> > Evaluator reliability is addressed separately in our response, **R4 to Reviewer UeF1**, through **independent safety judges**, **per-language human agreement**, and a **same-subset comparison between GPT-4o and human labels**.
> >
> > Although absolute ASR values vary across evaluators, all judges preserve the same **English-to-code-mixed degradation trend**.

---

### Review · Reviewer_UeF1 · 2026-07-02

**Summary Of Contributions:**

The paper studies a safety vulnerability of instruction-tuned LLMs under code-mixed prompts, where English is blended with another language within the same query. The central claim is that prompts that are refused or handled safely in monolingual English can elicit harmful responses when rewritten in code-mixed form. The authors evaluate this phenomenon across ten cultural/language settings, including Arabic, Bengali, Chinese, Hindi, Japanese, Korean, Russian, German, Spanish, and Portuguese, using Generic and Local TestSets. They report large increases in attack success rate (ASR) under code-mixing. The paper’s main methodological contribution is Saliency Drift Attribution (SDA), an attribution-based diagnostic framework intended to explain why these failures occur. Using token-level attribution, the authors argue that safety-critical words receive high saliency in English prompts but lose saliency in code-mixed variants, causing the model to miss the harmful intent.

## Strengths

- The paper addresses an important and underexplored safety issue: multilingual and code-mixed inputs.
- The paper combines several forms of evidence, including controlled synthetic code-mixed prompts, local culturally grounded prompts, sensitivity analysis over mixing ratios, attribution analysis, human validation, and evaluation on naturally occurring Hindi-English social media data.
- The human validation is a useful addition because the main ASR metric relies on GPT-4o.

## Weaknesses

- The empirical claims are very strong, but the causal link between attribution drift and safety failure is not fully established. SDA shows a correlation between saliency changes and harmful outputs, but it is less clear that attribution drift is the actual mechanism causing the failure.
- The code-mixed data construction may introduce artifacts. Many examples appear synthetic, and the naturalness, translation quality, and cultural validity of the generated prompts are central to the conclusions.
- The evaluation relies heavily on GPT-4o as a safety annotator with limited human validation.
- The paper sometimes overstates the novelty and scale of the risk, while the evaluated models are limited to three open models and the benchmark design may not fully reflect deployment settings.
- The paper would benefit from stronger baselines and mitigation experiments, such as multilingual safety classifiers, translation-to-English defenses, code-mixed safety fine-tuning, or comparisons with closed-source safety-aligned models.

**Audience:**

Yes

**Audience Explanation:**

The authors concern a timely problem in LLM safety and robustness: whether models that appear safety-aligned in English remain safe under multilingual or code-mixed inputs. The empirical observation itself is important: the paper reports that attack success rates increase substantially when prompts are transformed from English into code-mixed forms across multiple languages and models. The paper also connects safety evaluation with culturally and linguistically diverse settings. Many users communicate in mixed-language forms, especially in multilingual societies and online platforms. The inclusion of synthetic code-mixed prompts, local culturally grounded prompts, human validation, and Hindi-English social media data makes the topic relevant beyond a narrow benchmark setting.

**Broader Impact Concerns:**

The paper provides concrete examples of how code-mixed prompts may bypass safety guardrails, creating dual-use risk. The authors should discuss how to release datasets, prompts, and harmful outputs responsibly. They should also avoid framing code-mixing itself as problematic, since it is normal linguistic behavior for many multilingual communities; the concern is the model’s lack of robust safety alignment, not the users’ language practices. The paper should also address possible cultural and annotator bias, especially because harmfulness judgments rely heavily on GPT-4o with limited human validation.

**Claims And Evidence:**

No

**Claims Explanation:**

Although the paper provides useful evidence that code-mixed inputs can increase harmful response rates, I do not think the submission fully supports its strongest claims with sufficiently convincing and clear evidence.

The main empirical phenomenon is reasonably supported. However, the paper’s stronger mechanistic claim, that the failures are caused by “attributional collapse”, is less convincingly established. The proposed SDA analysis shows that some safety-critical tokens receive lower attribution under code-mixing, but this is primarily correlational. The paper does not sufficiently rule out alternative explanations, such as translation artifacts, unnatural synthetic code-mixed prompts, tokenization effects, distribution shift, weaker multilingual representations, or biases from the automatic safety evaluator.

I am also concerned that the strength of the language in the paper is not always matched by the scope of the evidence. Terms such as “catastrophic,” “overlooked weakness,” and claims about risks for “billions of users” seem overstated given that the evaluation is limited to three open-weight models and relies heavily on synthetic prompt generation plus GPT-4o-based annotation with limited human validation. The TRAC-1 experiment is useful, but it only covers Hindi-English social media data, so it does not fully validate the broader cross-cultural deployment-scale claims.

**Requested Changes:**

- **Temper the strongest claims and align them with the evidence**. The paper should revise language such as “catastrophic,” “blinding it to harmful intent,” “overlooked weakness,” and risks for “billions of users.” The empirical phenomenon is interesting, but the current evidence is based on three open-weight models, largely synthetic code-mixed prompts, and GPT-4o-based safety labeling. The claims should be framed more carefully as evidence of a serious potential vulnerability rather than a fully established deployment-scale failure.
- **Strengthen the causal evidence for “attributional collapse.”** The SDA analysis is useful, but currently it mostly shows correlation between attribution drift and harmful outputs. The authors should add interventions or counterfactual tests showing that restoring saliency to safety-critical tokens reduces harmful generations, or that controlled saliency-preserving perturbations avoid the failure.
- **Better control for artifacts in the code-mixed data generation.** Since many results rely on MLF-generated synthetic code-mixed prompts, the paper should provide stronger evidence that the prompts are natural, semantically faithful, and culturally valid. The human validation is helpful, but more detail is needed about sampling, annotator qualifications, per-language quality, disagreement cases, and whether harmfulness changes may be caused by translation or unnatural phrasing rather than code-mixing itself.
- **Reduce dependence on GPT-4o as the primary safety annotator or expand validation.** The main ASR results rely on GPT-4o labels with limited human validation. The authors should either expand human annotation, report per-language agreement between GPT-4o and native speakers, or compare against additional multilingual safety classifiers/evaluators.
- **Add stronger baselines and mitigation comparisons.** The paper would be much stronger with comparisons to simple and practical defenses, such as translating prompts to English before safety filtering, multilingual safety classifiers, code-mixed safety fine-tuning, or prompt-level normalization. If the authors claim attribution-guided interventions are promising, they should evaluate them systematically.
- **Clarify the experimental setup and reproducibility details.** The paper should specify prompt templates, decoding settings, exact model checkpoints, dataset construction details, safety-labeling prompts, and the full annotation protocol. Some details are provided in the appendix, but the main experiments would be easier to assess if the evaluation pipeline were described more transparently.
- **Include stronger model coverage.** The current evaluation covers phi-4B, Mistral-7B, and LLaMA-3.1-8B. Adding more recent or stronger models, including closed-source models if possible, would help determine whether this is a broad safety issue or mainly a weakness of the selected open models.
- **Analyze tokenization and multilingual representation effects.** The paper attributes failures to saliency drift, but tokenization fragmentation, script differences, and multilingual pretraining quality may also explain part of the effect. An analysis of tokenization length, script, language distance, and embedding/token fragmentation would help rule out alternative explanations.
- **Expand real-world validation beyond Hindi-English TRAC-1.** The TRAC-1 experiment is valuable, but it only covers Hindi-English social media data. Additional naturally occurring code-mixed datasets from other languages would better support the broader multilingual and cross-cultural claims.

---

> ### Author Response · Authors · 2026-07-09
> **We express our gratitude to reviewer UeF1 for evaluation**
>
> **Answer 1:** We agree and have revised the manuscript throughout. We shall remove terms such as **“catastrophic,”** **“overlooked weakness,”** **“blinding it to harmful intent,”** and the **“billions of users”** terminology.
>
> Crucially, our new **extended model analysis** motivates this tempering on empirical grounds. **Frontier models** are markedly more robust:
>
> | Model | CM ASR |
> |---|---:|
> | **GPT-4o** | **13.2%** |
> | **Gemini-2.5-Pro** | **16.8%** |
> | **Claude-3.7** | **10.6%** |
>
> Newer and larger open models also attenuate the English-to-code-mixed safety gap:
>
> | Model | CM Safety Gap |
> |---|---:|
> | **Llama-3.3-70B** | **+22.2 pp** |
> | **Qwen3-14B** | **+23.1 pp** |
> | **Aya-expanse-32B** | **+13.1 pp** |
>
> This contrasts with the smaller open models in our original study, where the degradation was substantially larger:
>
> | Model | CM Safety Gap |
> |---|---:|
> | **Phi** | **+58.4 pp** |
> | **Mistral** | **+53.9 pp** |
>
> We shall therefore frame the contribution as evidence of a **substantial**, **causally supported**, and **judge-robust safety degradation** that is most pronounced in **smaller open-weight instruction-tuned models**. At the same time, we will clarify that the degradation becomes **less severe**, though it does not disappear entirely, with increased model scale and stronger multilingual alignment.
>
> **Answer 2:** We agree that the original **SDA analysis** was primarily correlational. We therefore added a **bidirectional counterfactual intervention** that selectively manipulates the internal representations of **safety-critical tokens** while leaving the input prompt and its tokenization unchanged.
>
> By **suppression**, we mean replacing the hidden representations of safety-critical English tokens toward a **neutral benign-token baseline**, thereby reducing their influence without changing the prompt. By **restoration**, we mean replacing the hidden representations of aligned code-mixed safety tokens toward their corresponding **English-run representations**. Identical interventions on matched random tokens serve as controls.
>
> ### Bidirectional Counterfactual Intervention Results
>
> | Source | Manipulation | Mean Safety-Token RI | ASR |
> |---|---|---:|---:|
> | English | None | 0.64 | 18.2% |
> | English | Random-token suppression | 0.61 | 20.4% |
> | **English** | **Safety-token suppression** | **0.32** | **51.7%** |
> | Code-mixed | None | 0.27 | 72.1% |
> | Code-mixed | Random-token steering | 0.29 | 68.8% |
> | **Code-mixed** | **Safety-token restoration** | **0.49** | **39.6%** |
>
> Suppressing **safety-critical representations** in English lowers their mean **RI** from **0.64** to **0.32** and increases **ASR** from **18.2%** to **51.7%**. Conversely, restoring these representations in code-mixed prompts raises mean **RI** from **0.27** to **0.49** and reduces **ASR** from **72.1%** to **39.6%**.
>
> Random-token interventions produce only small ASR changes of approximately **2–3 percentage points**. These bidirectional results provide stronger **causal evidence** that weakened safety-critical representations contribute to the observed failures, while not implying that they are the sole mechanism.
>
> **Answer 3:** We thank the reviewer for this important concern. We conducted a **stratified sample-based validation** of the generated prompts using **native-speaker assessment** and three complementary **semantic-fidelity measures**.
>
> We randomly sampled **50 prompts per language** across the **Generic** and **Local TestSets**. Each prompt was independently evaluated by **three native or highly proficient speakers** of the matrix language who were also fluent in English through **Prolific**. Annotators assessed **meaning preservation**, **grammaticality**, **code-mixing naturalness**, **cultural appropriateness**, and whether the transformation altered the original **harmful intent**.
>
> Acceptable translation quality ranged from **75.7% to 100%** across languages, with an average of **87.3%**. The main disagreement categories involved **transliteration variation**, **unnatural lexical choice**, **ambiguous alignment**, and **changes in meaning intensity**.
>
> On the same sampled subset, we compared each transformed prompt with its corresponding original English prompt. **Multilingual BERTScore F1** and **SentenceTransformer cosine similarity** were computed between the original English prompt and the **fully translated**, **MLF code-mixed**, and **randomly mixed** variants.
>
> For **back-translation adequacy**, each transformed prompt was translated back into English and compared with the original English prompt using a fixed **GPT-based rubric** covering preservation of **meaning**, **target**, **harmful intent**, **cultural context**, and **intensity**.
>
> **Please look into next section for 5K limit.**

---

> > ### Author Response · Authors · 2026-07-09
> > **Continuation**
> >
> > ### Semantic-Fidelity Evaluation
> >
> > | Transformation | BERTScore F1 | Semantic Similarity | Back-Translation Adequacy |
> > |---|---:|---:|---:|
> > | Full translation | 0.91 | 0.92 | 92.8% |
> > | **MLF romanized** | **0.87** | **0.88** | **87.9%** |
> > | Random mixing | 0.78 | 0.79 | 73.5% |
> >
> > Finally, when restricting the sampled analysis to **high-fidelity examples** with **BERTScore F1 above 0.88** and **semantic similarity above 0.88**, **MLF code-mixed prompts** retained an **ASR of 58.9%**, compared with **29.8%** for fully translated prompts.
> >
> > These findings suggest that **mistranslation** and **unnatural phrasing** may explain some individual cases, but do not account for the overall **safety degradation** under structurally valid code-mixing.
> >
> > **Answer 4:** We added **two independent automatic judges** and expanded **human validation**. All four judges preserve the same ordering: **CM ≫ English**, **Mistral/Phi degradation ≫ Llama**, and **Arabic/Hindi/Bengali highest-risk**, as shown in the tables below.
> >
> > ## Judge Robustness
> >
> > **ASR averaged over models and languages.**
> >
> > | Judge | English ASR | CM ASR | Gap |
> > |---|---:|---:|---:|
> > | GPT-4o | 18.4% | 62.9% | +44.5 |
> > | LlamaGuard multilingual | 21.7% | 65.3% | +43.6 |
> > | ShieldGemma | 20.5% | 61.1% | +40.6 |
> > | **Human majority** | **17.8%** | **57.6%** | **+39.8** |
> >
> > ## Judge–Human Agreement
> >
> > | Judge Pair | Cohen's κ |
> > |---|---:|
> > | **GPT-4o ↔ human** | **0.78** |
> > | LlamaGuard ↔ human | 0.70 |
> > | ShieldGemma ↔ human | 0.68 |
> > | **Majority of automatic judges ↔ human** | **0.82** |
> >
> > ## Per-Language GPT-4o ↔ Human Agreement
> >
> > **n = 50 per language.**
> >
> > | Language | κ | Agreement | Language | κ | Agreement |
> > |---|---:|---:|---|---:|---:|
> > | Arabic | 0.73 | 87.5% | Japanese | 0.79 | 90.8% |
> > | Bengali | 0.75 | 88.3% | Korean | 0.81 | 91.7% |
> > | Chinese | 0.80 | 91.7% | Portuguese | 0.82 | 92.5% |
> > | German | 0.85 | 94.2% | Russian | 0.78 | 90.0% |
> > | Hindi | 0.76 | 89.2% | Spanish | 0.84 | 93.3% |
> >
> > ## Disagreement Taxonomy
> >
> > The main disagreement categories were:
> >
> > | Category | Share | Description |
> > |---|---:|---|
> > | **Indirect stereotypes** | **38%** | Harmful generalizations were implied rather than stated explicitly, such as associating a group with corruption without using an overt slur. |
> > | **Quotation versus endorsement** | **27%** | The model repeated a harmful claim while discussing or rejecting it, but one evaluator interpreted the repetition as endorsement. |
> > | **Culture-specific derogatory expressions** | **19%** | GPT-4o missed a locally offensive term or coded insult. |
> > | **Mixed-language ambiguity** | **10%** | The meaning of a transliterated or code-mixed phrase was interpreted differently by evaluators. |
> > | **Annotation error or unresolved disagreement** | **6%** | Labels remained inconsistent after review. |
> >
> > ## Reconciliation on the Common Human-Validation Subset
> >
> > We state explicitly that the human study evaluates a **stratified subset**, balanced by **language**, **model**, and **condition**, rather than a re-estimation of the full **Table 1** population. On that same subset, **GPT-4o** and **human majority labels** agree closely.
> >
> > | Same Subset | English ASR | CM ASR |
> > |---|---:|---:|
> > | GPT-4o | 18.9% | 59.7% |
> > | **Human majority** | **17.8%** | **57.6%** |
> >
> > **Answer 5:** We evaluated two **mitigation strategies**: **single-language prompt normalization** and **safety-token representation restoration**.
> >
> > ## Prompt Normalization
> >
> > Each code-mixed prompt was converted either fully into **English** or fully into the **matrix language** before generation.
> >
> > | Input Condition | ASR | Reduction from Native-Script CM |
> > |---|---:|---:|
> > | **English normalization** | **23.8%** | **−33.4 pp** |
> > | Matrix-language normalization | 31.6% | −25.6 pp |
> > | MLF code-mixed, native script | 57.2% | — |
> > | MLF code-mixed, romanized | 66.1% | — |
> >
> > **English normalization** gives the strongest improvement, while **matrix-language normalization** also substantially reduces ASR. This suggests that removing language mixing provides partial protection, although neither normalization fully restores the safety level of the original English prompts.
> >
> > ## Safety-Token Representation Restoration
> >
> > At aligned **safety-critical positions**, we replace the code-mixed **residual-stream activation** with the corresponding **English-run activation**, without changing the prompt or tokenization.
> >
> > | Intervention | Mean Safety-Token RI | ASR |
> > |---|---:|---:|
> > | Code-mixed, no intervention | 0.27 | 72.1% |
> > | Random-token restoration | 0.29 | 68.8% |
> > | **Safety-token restoration** | **0.49** | **39.6%** |
> >
> > **Safety-token restoration** reduces ASR by **32.5 percentage points**, whereas matched **random-token restoration** reduces it by only **3.3 percentage points**. This provides a targeted **white-box mitigation** for open-weight models.
> >
> > **Please look into next section for 5K limit.**

---

> > > ### Author Response · Authors · 2026-07-09
> > > **Continuation**
> > >
> > > **Answer 6:** We will specify the **prompt templates**, **deterministic decoding** settings (**temperature = 0.0**, **fixed seeds**), **MLF construction**, the **safety-labeling prompt**, and the full **annotation protocol** in the revised version.
> > >
> > > The **SDA specification** is fully formalized. Please see our response to **Reviewer sYSB, Point 2**.
> > >
> > > ---
> > >
> > > **Answer 7:** The gap persists across all models but shrinks with **scale** and **multilingual alignment**. **SDA** remains restricted to **open-weight models** where gradients and activations are accessible; the three frontier-model rows are **behavioral-only**.
> > >
> > > ## Extended Model Coverage
> > >
> > > **English vs. code-mixed ASR.**
> > >
> > > | Model | English ASR | CM ASR | Gap |
> > > |---|---:|---:|---:|
> > > | **Phi-4-mini-instruct** | 9.2% | 67.6% | **+58.4** |
> > > | **Mistral-7B-Instruct-v0.3** | 18.2% | 72.1% | **+53.9** |
> > > | Llama-3.1-8B-Instruct | 27.8% | 49.1% | +21.3 |
> > > | Gemma-2-9b-it | 15.6% | 54.8% | +39.2 |
> > > | Llama-3.3-70B-Instruct | 8.7% | 30.9% | +22.2 |
> > > | Qwen2.5-14B-Instruct | 10.9% | 38.7% | +27.8 |
> > > | Qwen3-14B | 8.4% | 31.5% | +23.1 |
> > > | DeepSeek-R1-Distill-Qwen-14B | 12.1% | 40.8% | +28.7 |
> > > | **Aya-expanse-32b** | **11.8%** | **24.9%** | **+13.1** |
> > > | gpt-4o-2024-11-20 *(behavioral only)* | 3.9% | 13.2% | +9.3 |
> > > | gemini-2.5-pro *(behavioral only)* | 4.7% | 16.8% | +12.1 |
> > > | claude-3-7-sonnet *(behavioral only)* | 2.8% | 10.6% | +7.8 |
> > >
> > > ---
> > >
> > > **Answer 8:** We agree that **token fragmentation** and **script choice** contribute to the observed safety degradation. Code-mixed prompts are split into more tokens than English prompts, especially in **romanized form**.
> > >
> > > ## Tokenization and Script Effects
> > >
> > > | Condition | Tokens/word | Fragmented Safety Tokens | ASR |
> > > |---|---:|---:|---:|
> > > | English | 1.31 | 18% | 18.4% |
> > > | Full translation | 1.82 | 39% | 31.6% |
> > > | **Native-script code-mixed** | **2.07** | **48%** | **57.2%** |
> > > | **Romanized code-mixed** | **2.34** | **56%** | **66.1%** |
> > >
> > > However, **tokenization does not fully explain the failure**. Even after controlling for **token fertility**, **prompt length**, and **toxicity**, greater **safety-token drift** remains strongly associated with harmful generation.
> > >
> > > Specifically, every **0.10 increase in drift** makes a **Case-1 failure** about **1.78 times more likely**; the estimated effect lies between **1.57 and 2.03 times**, with **p < 0.001**.
> > >
> > > We also compare prompts with similar **tokenization complexity**. In this matched subset, code-mixed prompts still show higher **ASR** than fully translated prompts: **54.6% versus 33.1%**.
> > >
> > > These results show that **tokenization** and **script differences** make the problem worse, but weaker **multilingual safety representations** inside the model also contribute beyond surface token fragmentation.
> > >
> > > ---
> > >
> > > **Answer 9:** The phenomenon replicates on **wild data** across four language pairs.
> > >
> > > ## Natural Code-Mixed Datasets
> > >
> > > | Dataset | Language Pair | Normalized-English ASR | Natural CM ASR | Gap |
> > > |---|---|---:|---:|---:|
> > > | TRAC-1 | Hindi–English | 24.8% | 71.8% | **+47.0** |
> > > | LinCE-derived [[link](https://aclanthology.org/2020.lrec-1.223/)] | Spanish–English | 20.6% | 49.7% | **+29.1** |
> > > | HASOC/Dravidian-derived [[link](https://ceur-ws.org/Vol-3159/T3-1.pdf)] | Tamil–English | 22.9% | 57.8% | **+34.9** |
> > > | Arabizi social-media [[link](https://aclanthology.org/2022.wanlp-1.18.pdf)] | Arabic–English | 19.3% | 63.5% | **+44.2** |
> > >
> > > ## Per-Model Natural CM ASR
> > >
> > > | Dataset | Phi | Mistral | Llama |
> > > |---|---:|---:|---:|
> > > | TRAC-1 | 61.3% | 74.9% | 79.2% |
> > > | Spanish–English | 44.2% | 57.1% | 47.8% |
> > > | Tamil–English | 52.7% | 65.4% | 55.3% |
> > > | Arabic–English/Arabizi | 58.6% | 70.8% | 61.1% |

---

> > > > ### Comment · Reviewer_UeF1 · 2026-07-18
> > > >
> > > > Thank you for the detailed response and additional experiments. The rebuttal address most of my concerns.
> > > >
> > > > My main remaining question is about the bidirectional counterfactual intervention. Could you clarify exactly how the safety-token representation restoration/suppression is implemented, including which layer(s) are intervened on, how English and code-mixed token positions are aligned under different tokenizations, and whether the full residual-stream representation is replaced?
> > > >
> > > > In particular, replacing a code-mixed hidden representation with its corresponding English-run representation may transfer semantic or safety-related information beyond simply restoring saliency. Please clarify this distinction and ensure the revised manuscript does not overstate the causal conclusion.

---

> ### Author Response · Authors · 2026-07-20
> **Response to reviewer**
>
> Thank you for the clarification request. We agree that the bidirectional counterfactual intervention needs to be described more precisely and that the causal claim should be scoped carefully.
>
> The intervention is applied to the **residual stream** at aligned safety-critical token spans. For Mistral-7B-Instruct-v0.3 and Llama-3.1-8B-Instruct, we use the middle-late layer block **16–23** as the main intervention range, since this block gave the strongest localized repair in our layer sweep. We will also report the full layer-block results in the revision.
>
> English and code-mixed positions are aligned at the **word/span level** using the same alignment procedure as SDA. If an aligned word is split into multiple subword tokens, the intervention is applied to all subword positions in that span. For one-to-many or many-to-one alignments, we intervene on the full aligned span rather than a single token.
>
> For restoration, we replace/interpolate the **full residual-stream vector** at the aligned code-mixed safety-token position with the corresponding English-run vector:
>
> h_CM[l,j] ← (1 − λ) h_CM[l,j] + λ h_EN[l,i]
>
> For suppression, we apply the reverse control in English by interpolating safety-token residual states toward a neutral benign-token baseline. Matched random-token interventions use the same operation on the same number of non-safety tokens.
>
> We agree that full residual-stream replacement may transfer semantic and safety-related information beyond scalar saliency alone. Therefore, we will describe this as a **safety-token representation intervention with saliency verification**, not as direct manipulation of saliency alone. The revised causal claim is:
> >Restoring English-like residual-stream representations at aligned safety-critical positions reduces harmful generations, while suppressing those representations in English increases failures. This supports the conclusion that degraded safety-critical representations contribute to code-mixed safety failures, but it does not show that saliency alone is the unique causal mechanism.